# Pancreatic cancer is associated with medication changes prior to clinical diagnosis

Yin Zhang [1,2], Qiao-Li Wang[1,3], Chen Yuan[1], Alice A. Lee [4], Ana Babic[1], Kimmie Ng [1], Kimberly Perez[1], Jonathan A. Nowak[5], Jesper Lagergren[6,7], Meir J. Stampfer[2,8,9], Edward L. Giovannucci[2,9], Chris Sander[10,11,12], Michael H. Rosenthal[13], Peter Kraft [9,14] & Brian M. Wolpin[1] ✉

Patients with pancreatic ductal adenocarcinoma (PDAC) commonly develop symptoms and signs in the 1–2 years before diagnosis that can result in changes to medications. We investigate recent medication changes and PDAC diagnosis in Nurses' Health Study (NHS; females) and Health Professionals Follow-up Study (HPFS; males), including up to 148,973 U.S. participants followed for 2,994,057 person-years and 991 incident PDAC cases. Here we show recent initiation of antidiabetic (NHS) or anticoagulant (NHS, HFS) medications and cessation of antihypertensive medications (NHS, HPFS) are associated with pancreatic cancer diagnosis in the next 2 years. Two-year PDAC risk increases as number of relevant medication changes increases ($P$-trend $<1 \times 10^{-5}$), with participants who recently start antidiabetic and stop antihypertensive medications having multivariable-adjusted hazard ratio of 4.86 (95%CI, 1.74–13.6). These changes are not associated with diagnosis of other digestive system cancers. Recent medication changes should be considered as candidate features in multi-factor risk models for PDAC, though they are not causally implicated in development of PDAC.

In the United States, pancreatic cancer is the 3rd leading cause of cancer-related death[1]. Approximately 80% of patients present with advanced disease, which is difficult to cure and is associated with short survival times[2]. Although earlier detection is a critical priority, average-risk general population screening is difficult due to the relatively low disease incidence (39 per 100,000 person-years for women and 50 per 100,000 person-years for men among those ≥50 years of age)[3,4]. The US Preventive Services Task Force (USPSTF), after weighing the potential benefits and harms, recently reaffirmed a recommendation against screening for pancreatic cancer in asymptomatic average-risk adults[5]. Identifying high-risk subsets who might benefit from surveillance is critical to pancreatic cancer early detection[6,7], but currently is

[1]Department of Medical Oncology, Dana-Farber Cancer Institute and Harvard Medical School, Boston, MA, USA. [2]Department of Nutrition, Harvard T. H. Chan School of Public Health, Boston, MA, USA. [3]Department of Clinical Science, Intervention and Technology, Karolinka Institutet, Stockholm, Sweden. [4]Division of Gastroenterology, Hepatology and Endoscopy, Brigham and Women's Hospital and Harvard Medical School, Boston, MA, USA. [5]Program in MPE Molecular Pathological Epidemiology, Department of Pathology, Brigham and Women's Hospital and Harvard Medical School, Boston, MA, USA. [6]Upper Gastrointestinal Surgery, Department of Molecular Medicine and Surgery, Karolinska Institutet, Karolinska University Hospital, Stockholm, Sweden. [7]School of Cancer and Pharmaceutical Sciences, King's College London, London, UK. [8]Channing Division of Network Medicine, Department of Medicine, Brigham and Women's Hospital and Harvard Medical School, Boston, MA, USA. [9]Department of Epidemiology, Harvard T. H. Chan School of Public Health, Boston, MA, USA. [10]Department of Data Science, Dana-Farber Cancer Institute, Boston, MA, USA. [11]Department of Cell Biology, Harvard Medical School, Boston, MA, USA. [12]Broad Institute of Harvard and MIT, Cambridge, MA, USA. [13]Department of Radiology, Brigham and Women's Hospital and Harvard Medical School, Boston, MA, USA. [14]Department of Biostatistics, Harvard T.H. Chan School of Public Health, Boston, MA, USA. ✉e-mail: brian_wolpin@dfci.harvard.edu

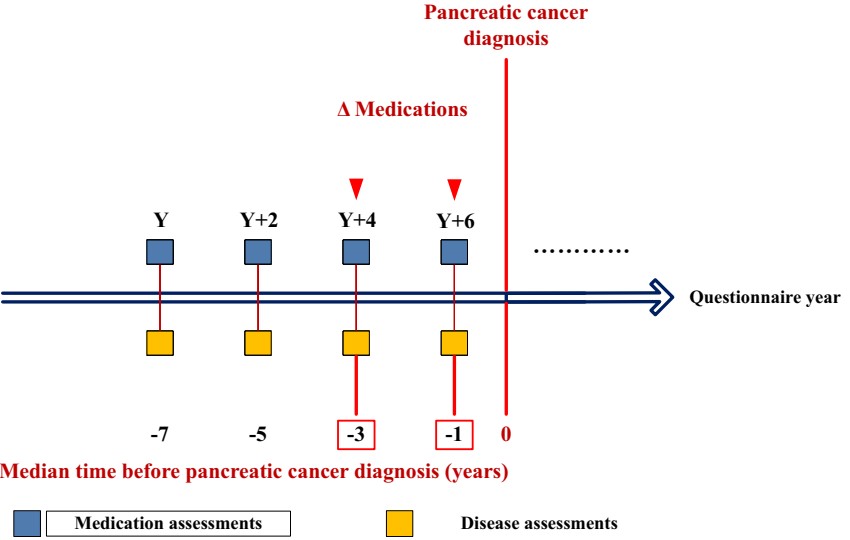

**Fig. 1 | Schematic of assessments of medication change in the prediagnosis time period for pancreatic cancer.** Δ: Medication change. Throughout follow-up, for survey cycles in which pancreatic cancer was diagnosed, medication changes in the prediagnosis time window were measured by comparing questionnaires returned at a median of 1 year (current status of medication use) and 3 years (previous status of medication use) before diagnosis. Y: Year the cohort study was initiated, with data collected every 2 years thereafter by mailed questionnaire.

pursued only among those with high familial or genetic risk and individuals with pancreatic cystic lesions[8,9]. Defining additional patient features for inclusion in multi-factor models that identify individuals at near-term high risk for pancreatic cancer is important to expanding the population appropriate for pancreatic cancer early detection.

In the several years before a clinical diagnosis of pancreatic cancer, patients develop symptoms and signs that might serve as an early warning for the growing malignancy. For example, new-onset metabolic alterations such as hyperglycemia and weight loss occur during this time window and are increasingly recognized as defining a high-risk group[4,10–12]. Similarly, pancreatic cancer patients are commonly noted in clinical practice to have started or stopped certain medications in the 1–2 years before their clinical diagnosis, due to development of new symptoms or conditions such as hyperglycemia, venous thromboembolism, abdominal discomfort, and depression[13,14]. Nevertheless, few studies have characterized the value of these medication changes in defining those at risk of a subsequent pancreatic cancer diagnosis[15,16].

To examine medication changes in the two-year prediagnosis time period, we investigated longitudinal, prospectively collected assessments of medication use among up to 148,973 participants from the Nurses' Health Study (NHS, a large prospective cohort of US female nurses)[17–20] and Health Professionals Follow-up Study (HPFS, a large prospective cohort of US male health professionals)[21], to determine the association of medication change patterns with subsequent pancreatic cancer development. Specifically, we hypothesized that starting antidiabetic, anticoagulant, antacid, non-steroidal anti-inflammatory drug (NSAID), or antidepressant medications and stopping antihypertensive medications would be associated with increased likelihood of pancreatic cancer development in the next 2 years. We also considered whether these changes were specific to patients with pancreatic cancer or more generally identified among individuals who developed other cancer types.

In this work, recent initiation of antidiabetic or anticoagulant medications and cessation of antihypertensive medications are associated with pancreatic cancer diagnosis in the next 2 years. The highest two-year risks for pancreatic cancer are identified among those with recent changes in 2 or more of antidiabetic, anticoagulant, and antihypertensive medications. In contrast, recent changes in these medications are not associated with development of other gastrointestinal cancers, suggesting specificity to development of pancreatic cancer. Our findings support that recent medication changes should be considered as candidate features in multi-factor models for pancreatic cancer early detection, though these medication changes are not causally implicated in development of pancreatic cancer.

## Results

Study schema and populations for medication use analyses are summarized in Fig. 1 and Supplementary Tables 1–3. Analyses of anticoagulant medications, antihypertensive medications, antacids, NSAIDs, and antidepressants were first conducted in the combined dataset of both cohorts (both sexes), and then in the NHS (women) and HPFS (men) separately. Given the short time interval with available data for antidiabetic medication use in HPFS (men), analyses of antidiabetic medications and analyses of the combined antidiabetic, anticoagulant, and antihypertensive medications were performed in the NHS cohort (women) only (Supplementary Table 1).

### Recent change in medication use and 2-year risk of pancreatic cancer diagnosis

We first assessed the association of changes in antidiabetic medication use and pancreatic cancer development in the next 2 years in the NHS cohort (588 incident pancreatic cancer cases), given the known relationship between recent-onset hyperglycemia and pancreatic cancer[4]. In unadjusted analyses, the start of an OHG medication (HR, 2.96; 95% CI, 1.93-4.53) or insulin (HR, 5.28; 95% CI, 3.05-9.15) were both associated with pancreatic cancer development when compared to those with no change in use of these medications (Table 1, Supplementary Table 4). Both obesity and diabetes are known risk factors for pancreatic cancer, so we also examined multivariable-adjusted models that included BMI and diabetes. In the fully adjusted models, initiation of OHG medications had a HR of 1.39 (95% CI, 0.86–2.27) and insulin had a HR of 2.84 (95% CI, 1.58–5.12) for developing pancreatic cancer. We next created a composite measure of starting either OHG or insulin, which demonstrated a HR for pancreatic cancer of 1.99 (95% CI, 1.30–3.03; Table 1, Supplementary Table 4). In contrast, those

**Table 1 | Recent change in medication use and 2-year risk of pancreatic cancer diagnosis**

| | Recent change in medication use[a,b] | | |
|---|---|---|---|
| | No change | Start | Stop |
| **Any antidiabetic medications** | | | |
| No. of cases | 538 | 33 | 17 |
| Person-years | 1,966,617 | 33,435 | 15,660 |
| Incidence rate (per 100,000 person-years) | 27 (25–30) | 99 (70–139) | 109 (67–175) |
| Unadjusted | 1 [Ref] | 3.61 (2.54–5.13) | 3.97 (2.45–6.43) |
| MV-adjusted [c] | 1 | 1.99 (1.30–3.03) | 1.55 (0.86–2.81) |
| **Insulin** | | | |
| No. of cases | 570 | 13 | 5 |
| Person-years | 2,002,769 | 8,653 | 4,290 |
| Incidence rate (per 100,000 person-years) | 28 (26–31) | 150 (87–259) | 117 (49–280) |
| Unadjusted | 1 | 5.28 (3.05–9.15) | 4.10 (1.70–9.88) |
| MV-adjusted [c] | 1 | 2.84 (1.58–5.12) | 1.18 (0.43–3.24) |
| **Anticoagulant medications** | | | |
| No. of cases | 658 | 20 | 10 |
| Person-years | 1,754,044 | 25,461 | 12,227 |
| Incidence rate (per 100,000 person-years) | 38 (35–40) | 79 (51–122) | 82 (44–152) |
| Unadjusted | 1 | 2.09 (1.34–3.27) | 2.18 (1.17–4.07) |
| MV-adjusted [c] | 1 | 1.50 (0.95–2.35) | 1.18 (0.57–2.46) |
| **Antihypertensive medications** | | | |
| No. of cases | 789 | 86 | 116 |
| Person-years | 2,627,927 | 224,273 | 141,857 |
| Incidence rate (per 100,000 person-years) | 30 (28–32) | 38 (31–47) | 82 (68–98) |
| Unadjusted | 1 | 1.28 (1.02–1.60) | 2.72 (2.24–3.31) |
| MV-adjusted [c] | 1 | 1.03 (0.81–1.30) | 1.77 (1.42–2.20) |
| **Antacids** | | | |
| No. of cases | 229 | 21 | 19 |
| Person-years | 539,960 | 62,898 | 45,475 |
| Incidence rate (per 100,000 person-years) | 42 (37–48) | 33 (22–51) | 42 (27–66) |
| Unadjusted | 1 | 0.79 (0.50–1.23) | 0.99 (0.62–1.57) |
| MV-adjusted [c] | 1 | 0.76 (0.48–1.21) | 1.06 (0.58–1.93) |
| **NSAIDs** | | | |
| No. of cases | 800 | 49 | 54 |
| Person-years | 2,333,006 | 148,689 | 129,881 |
| Incidence rate (per 100,000 person-years) | 34 (32–37) | 33 (25–44) | 42 (32–54) |
| Unadjusted | 1 | 0.96 (0.72–1.28) | 1.21 (0.92–1.60) |
| MV-adjusted [c] | 1 | 1.04 (0.76–1.42) | 1.14 (0.85–1.52) |
| **Antidepressants** | | | |
| No. of cases | 707 | 20 | 28 |
| Person-years | 1,790,806 | 51,326 | 38,441 |
| Incidence rate (per 100,000 person-years) | 39 (37–42) | 39 (25–60) | 73 (50–105) |
| Unadjusted | 1 | 0.99 (0.63–1.54) | 1.84 (1.26–2.69) |
| MV-adjusted [c] | 1 | 0.99 (0.63–1.55) | 1.39 (0.85–2.27) |

[a] Follow-up time: antidiabetic medications (NHS: 1990–2012; information on antidiabetic medication use in HPFS was assessed relatively late (in 2008 and afterwards) and was therefore not included in the analyses), anticoagulant (NHS: 1996-2012; HPFS: 1998-2012), antihypertensive medications (NHS: 1990–2012; HPFS: 1988–2012), antacids (NHS: 2002–2012; HPFS: 2006–2012), NSAIDs (NHS: 1992–2012; HPFS: 1988–2012), and antidepressants (NHS: 1998–2012; HPFS: 1992–2012).
[b] Medication change was measured by comparing questionnaires returned at a median of 1 year (current status of use) and 3 years (previous status of use) before pancreatic cancer diagnosis.
[c] Stratified by age (in months), sex/cohort (women, men; in the pooled analyses only), and calendar year of the survey cycle (each 2-year interval); adjusted for race/ethnicity (white, black, other, unknown), BMI (continuous, kg/m$^2$), physical activity (continuous, MET-hours/week), smoking (continuous, pack-years), alcohol intake (continuous, grams/day), history of diabetes (yes, no), multivitamin use (yes, no), and previous status of use (yes, no).
Abbreviations: *NHS* Nurses' Health Study, *HPFS* Health Professionals Follow-up Study, *MV* multivariate, *BMI* body mass index, *MET* metabolic equivalent task, *NSAIDs* non-steroidal anti-inflammatory drugs.

individuals who stopped antidiabetic medications were not more likely to develop pancreatic cancer (Table 1, Supplementary Table 4).

We next considered initiation of anticoagulant medication in the combined NHS and HPFS cohorts (688 incident pancreatic cancer cases), as venous thromboembolism is a known complication for patients with pancreatic cancer[14,22]. In multivariable-adjusted models, we noted a suggestively increased risk of pancreatic cancer in the next 2 years after starting an anticoagulant medication, with HR of 1.50 (95% CI, 0.95–2.35) comparing those who started to no change in anticoagulant medication use (Table 1, Supplementary Table 4). In contrast, cessation of anticoagulant medication was not associated with future pancreatic cancer (HR, 1.18; 95% CI, 0.57–2.46) (Table 1, Supplementary Table 4). Patients commonly lose weight and suffer metabolic derangements prior to their pancreatic cancer diagnosis[10–12,14,23,24]. Clinically, this can manifest as a reduction in blood pressure, which may reduce the need for antihypertensive medications. In fully adjusted analyses of changes in antihypertensive medication use in the combined NHS and HPFS cohorts (991 incident pancreatic cancer cases), we observed that those individuals who recently stopped antihypertensive medication were at higher risk for pancreatic cancer in the next 2 years (HR, 1.77; 95% CI, 1.42–2.20), whereas starting antihypertensive medication was not associated with subsequent pancreatic cancer (Table 1, Supplementary Table 4).

We considered three other medication classes for which medication initiation might be indicative of a subsequent pancreatic cancer diagnosis, including antacids in the combined NHS and HPFS cohorts (269 incident pancreatic cancer cases)[13–15], NSAIDs in the combined NHS and HPFS cohorts (903 incident pancreatic cancer cases)[13,14], and antidepressant medications in the combined NHS and HPFS cohorts (755 incident pancreatic cancer cases)[13,14]. Changes in use of these additional medication classes were not associated with 2-year pancreatic cancer risk (Table 1, Supplementary Table 5).

We conducted sensitivity analyses in which the referent group was considered those with stable non-use of a medication class and the three exposure categories included stable medication use, medication start, and medication stop (Supplementary Tables 6–7). Overall, results were highly similar, with increased 2-year pancreatic cancer risk among those individuals who started antidiabetic and anticoagulant medications or stopped antihypertensive medications. We also considered results among the separate study populations for each cohort, and results were similar, although the observed associations were stronger in the NHS where more incident pancreatic cancer cases were available for analyses (Supplementary Tables 8–11). In sensitivity analyses replacing BMI with recent weight change in multivariable-adjusted models, results remained highly similar (Supplementary Tables 4–11).

## Combined associations of recent changes in medication use with 2-year risk of pancreatic cancer diagnosis

We next investigated the combined associations of changes in antidiabetic, anticoagulant, and antihypertensive medication with development of pancreatic cancer in the NHS cohort (376 incident pancreatic cancer cases). Pancreatic cancer risk increased as the number of relevant medication changes increased (*P*-trend $<1 \times 10^{-5}$) (Fig. 2, Supplementary Table 12). We further explored the combined associations of changes in antidiabetic and antihypertensive medication use in the NHS cohort (588 incident pancreatic cancer cases). In multivariable-adjusted analyses, compared with participants who had no change in use of both medications, those who reported starting antidiabetic and stopping antihypertensive medication experienced the highest 2-year risk of developing pancreatic cancer (HR, 4.86; 95% CI, 1.74–13.6) (Table 2). Results were similar in propensity score-adjusted analyses (Table 2).

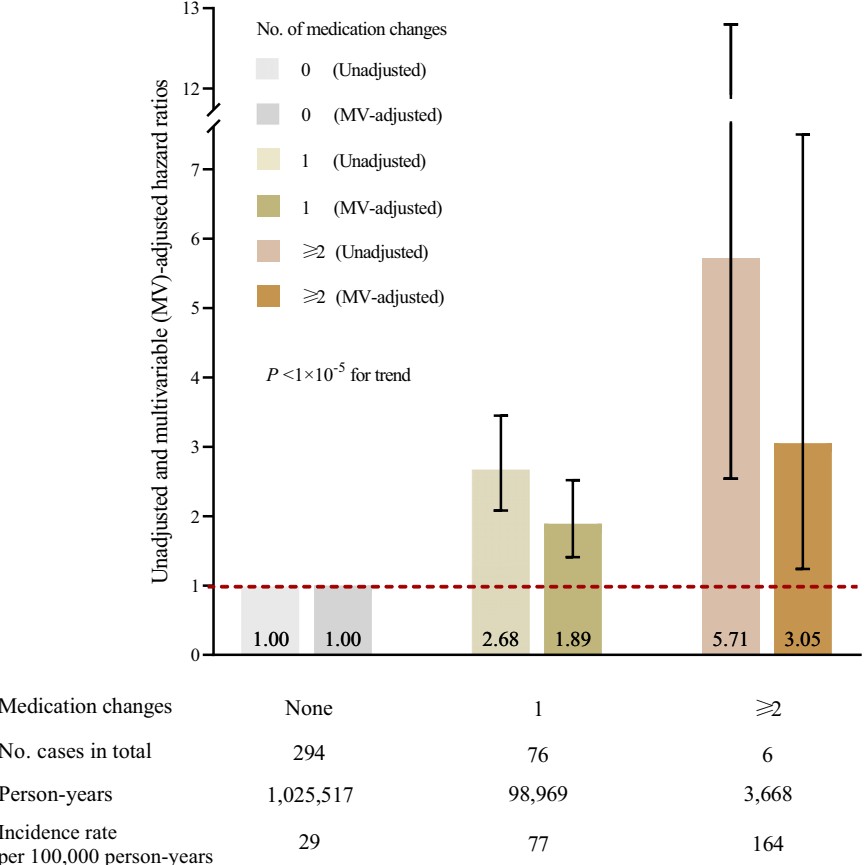

| Medication changes | None | 1 | ≥2 |
|---|---|---|---|
| No. cases in total | 294 | 76 | 6 |
| Person-years | 1,025,517 | 98,969 | 3,668 |
| Incidence rate per 100,000 person-years | 29 | 77 | 164 |

**Fig. 2 | Combined associations of recent changes in medication use with 2-year risk of pancreatic cancer diagnosis.** Medication changes include the start of antidiabetic or anticoagulant medications and the stop of antihypertensive medications. Data are presented as the hazard ratios obtained from Cox proportional hazards models in analyses of combined antidiabetic, anticoagulant, and antihypertensive medication changes (*n* = 84,623 biologically independent participants, 376 incident pancreatic cancer cases). Error bars indicate 95% confidence intervals. Multivariable-adjusted hazard ratios were stratified by age (in months) and calendar year of the survey cycle (each 2-year interval); and adjusted for race/ ethnicity (white, black, other, unknown), BMI (continuous, kg/m$^2$), physical activity (continuous, MET-hours/week), smoking (continuous, pack-years), alcohol intake (continuous, grams/day), history of diabetes (yes, no), multivitamin use (yes, no), and previous status of use (yes, no). Two-sided *P*-value for trend was calculated by entering the number of medication changes as an ordinal variable and assessed by the Wald test without correction for multiple comparisons. *P*-value <1 × 10$^{-5}$ for trend in MV-adjusted analysis. Abbreviations: MV multivariable, BMI body mass index, MET metabolic equivalent task.

## Specificity of medication Changes for Subsequent Pancreatic Cancer Diagnosis

We then considered whether medication changes associated with 2-year risk of pancreatic cancer diagnosis were associated with other cancer types. Given similarities in histology, risk factors, and presentation, we focused on other cancers of the digestive system, including cancers of esophagus, stomach, small intestine, colorectum, anus, liver, and gallbladder. Notably, changes in these medications were not associated with the risk of developing other digestive system cancers. (Fig. 3, Supplementary Table 13).

## Discussion

In this large prospective study, we explored the association between medication change in the prediagnosis time window and pancreatic cancer development. We hypothesized that medication changes would be reported by participants before their pancreatic cancer was diagnosed clinically due to early symptoms and systemic consequences from the undiagnosed cancer. Furthermore, we hypothesized that these medication changes would help risk stratify individuals even when other known risk factors were considered, while conferring a degree of specificity for pancreatic cancer compared to other cancer types due to the unique symptomatology and metabolic consequences of this malignancy. Indeed, changes in use for several

medications were associated with increased two-year risk of pancreatic cancer diagnosis after accounting for other risk factors, suggesting that recent changes in health status may provide additive risk stratification information. Since few conditions are known to portend sufficiently high pancreatic cancer risk to justify imaging surveillance[2], the identification of such further predictive features may allow construction of multi-factor models with sufficient predictive capacity for clinical utility[25-28]. It should be emphasized that the premise of the current study was that pancreatic cancer would cause symptoms or signs that lead to medication changes in the 1–2 years before diagnosis, rather than these medication changes were causally responsible for the development of pancreatic cancer.

In modern electronic medical record (EMR) systems, the provision of handwritten prescriptions from a prescription pad has been largely replaced by digital prescriptions that are typed into the EMR[29-31]. This change in medication prescriptions may reduce medical errors while increasing patient convenience[29,30,32-34]. Another potential benefit of this approach is that prescriptions become searchable structured fields within the EMR[30,31]. In fact, changes to medication prescribing can be a highly efficient way to assess new disease states from structured data[35,36], avoiding complexities associated with billing codes and free text notes. Furthermore, changes in medication use can inform disease severity over time[37,38], providing more information than

**Table 2 | Combined associations of recent changes in antidiabetic and antihypertensive medication use with 2-year risk of pancreatic cancer diagnosis**

| Medication use [a,b] | No. cases | Person-years | Incidence rate (per 100,000 person-years) | Hazard ratios (95% CIs) | | |
|---|---|---|---|---|---|---|
| | | | | Unadjusted | MV-adjusted [c] | Propensity score-adjusted [d] |
| No change in antidiabetic | | | | | | |
| + No change in antihypertensive | 427 | 1,735,359 | 25 (22–27) | 1 [Ref] | 1 | 1 |
| + Start antihypertensive | 49 | 145,087 | 34 (26–45) | 1.37 (1.02–1.84) | 1.16 (0.85–1.58) | 1.16 (0.85–1.59) |
| + Stop antihypertensive | 62 | 86,170 | 72 (56–92) | 2.92 (2.24–3.82) | 2.00 (1.49–2.69) | 2.01 (1.50–2.70) |
| Start antidiabetic | | | | | | |
| + No change in antihypertensive | 20 | 23,094 | 87 (56–134) | 3.52 (2.25–5.51) | 1.88 (1.13–3.12) | 1.84 (1.10–3.06) |
| + Start antihypertensive | 9 | 8,824 | 102 (53–196) | 4.15 (2.14–8.02) | 2.46 (1.20–5.06) | 2.52 (1.22–5.19) |
| + Stop antihypertensive | 4 | 1,517 | 264 (99–703) | 10.7 (4.00–28.7) | 4.86 (1.74–13.6) | 4.63 (1.65–13.0) |
| Stop antidiabetic | | | | | | |
| + No change in antihypertensive | 7 | 9,081 | 77 (37–162) | 3.13 (1.48–6.61) | 1.20 (0.52–2.73) | 1.22 (0.53–2.78) |
| + Start antihypertensive | – | – | – | – | – | – |
| + Stop antihypertensive | 10 | 5,816 | 172 (93-320) | 6.99 (3.73–13.1) | 2.58 (1.26–5.27) | 2.55 (1.25–5.22) |

[a] Follow-up time: antidiabetic medications (NHS: 1990–2012; information on antidiabetic medication use in HPFS was assessed relatively late (in 2008 and afterwards) and was therefore not included in the analyses), and antihypertensive medications (NHS: 1990–2012).
[b] Medication change was measured by comparing questionnaires returned at a median of 1 year (current status of use) and 3 years (previous status of use) before pancreatic cancer diagnosis.
[c] Stratified by age (in months), sex/cohort (women, men; in the pooled analyses only), and calendar year of the survey cycle (each 2-year interval); adjusted for race/ethnicity (white, black, other, unknown), BMI (continuous, kg/m²), physical activity (continuous, MET-hours/week), smoking (continuous, pack-years), alcohol intake (continuous, grams/day), history of diabetes (yes, no), multivitamin use (yes, no), and previous status of use (yes, no).
[d] Adjusted for propensity score.
Abbreviations: *NHS* Nurses' Health Study, *HPFS* Health Professionals Follow-up Study, *MV* multivariate, *BMI* body mass index, *MET* metabolic equivalent task.

the simple presence of absence of a condition. Given the increasing availability of structured data on medication use in the EMR, we conducted the current study as a proof of principle examination of whether changes in medication use inform 2-year risk of a pancreatic cancer diagnosis.

Diabetes is both a long-term risk factor for pancreatic cancer and is caused by pancreatic cancer in the several years prior to cancer diagnosis. Thus, altered glucose homeostasis is one of the most prevalent prediagnostic phenotypic traits in patients who develop pancreatic cancer[4,6,14,15,39,40]. Importantly, new-onset diabetes and worsening of glucose control in long-standing diabetes may occur in up to half of all patients who develop pancreatic cancer[10,12,39,41,42]. Rapid increases in blood glucose are also more frequent in patients with pancreatic cancer-associated diabetes compared to type 2 diabetes in the general population[43]. Thus, the addition of antidiabetic medications to a patient's medication program may signal worsening glucose control[16], providing risk stratifying information in both those with long-term and recent-onset diabetes and even if those conditions are known from billing codes or provider notes. In the current study, initiation of any antidiabetic medication was associated with a nearly 2-fold increased risk of pancreatic cancer in the next 2 years in females, even when accounting for a patient's BMI, recent weight loss, or prior diabetes diagnosis. Furthermore, stronger associations were identified for insulin compared to oral hypoglycemic medications, indicating the utility of medication change to provide insights into disease severity.

We also identified increased two-year risks of pancreatic cancer diagnosis for individuals who started anticoagulant or stopped antihypertensive medications. Pancreatic cancer is known to induce a hypercoagulable state, which can lead to venous thromboembolism[14,22]. These blood clots are treated with systemic anticoagulation[44]. Thus, initiation of these medications may serve as a straightforward approach to identifying individuals with new hypercoagulability that may develop prior to the clinical diagnosis of

pancreatic cancer. Patients with pancreatic cancer often experience weight loss in the time period prior to cancer diagnosis. In fact, hyperglycemia and weight loss commonly co-occur in patients who develop pancreatic cancer[4,43]. We hypothesized that as a consequence of weight loss and altered metabolism, blood pressure may decline in the prediagnosis period, no longer necessitating use of antihypertensive medications. Indeed, after accounting for patients' BMI, recent weight loss, and prior diabetes diagnosis, we identified a nearly 2-fold increased 2-year risk of pancreatic cancer diagnosis after cessation of antihypertensive medications. Interestingly, none of these medication changes were associated with development of other gastrointestinal cancers. Thus, the somewhat unique pathophysiology of pancreatic cancer may provide utility in earlier cancer detection, even when considered in terms of specificity for pancreatic cancer compared to other malignancies[11].

We identified no evidence supporting the hypothesized associations of starting antacids, NSAIDs or antidepressants with subsequent risk of pancreatic cancer, although several related clinical patterns before diagnosis (e.g., epigastric or back discomfort, new-onset gastroesophageal reflux, and depression) have been observed in clinical care[13,14]. As medication change was determined on a questionnaire at a median of 12 months prior to cancer diagnosis, it is possible these symptoms occurred too close to the clinical diagnosis for us to detect the medication change.

Our study has several important strengths. We simultaneously examined several types of medications in two prospective studies with up to 148,973 eligible participants and 24 years of follow-up. The prospective study design with enrollment prior to cancer diagnosis in conjunction with very low rates of loss to follow-up minimized the potential for exposure misclassification, recall bias and issues with patient selection. Validated time-varying information on a wide spectrum of covariates permitted rigorous control for known or plausible confounding. Specificity of our findings for pancreatic cancer compared to other malignancies was examined within the same large

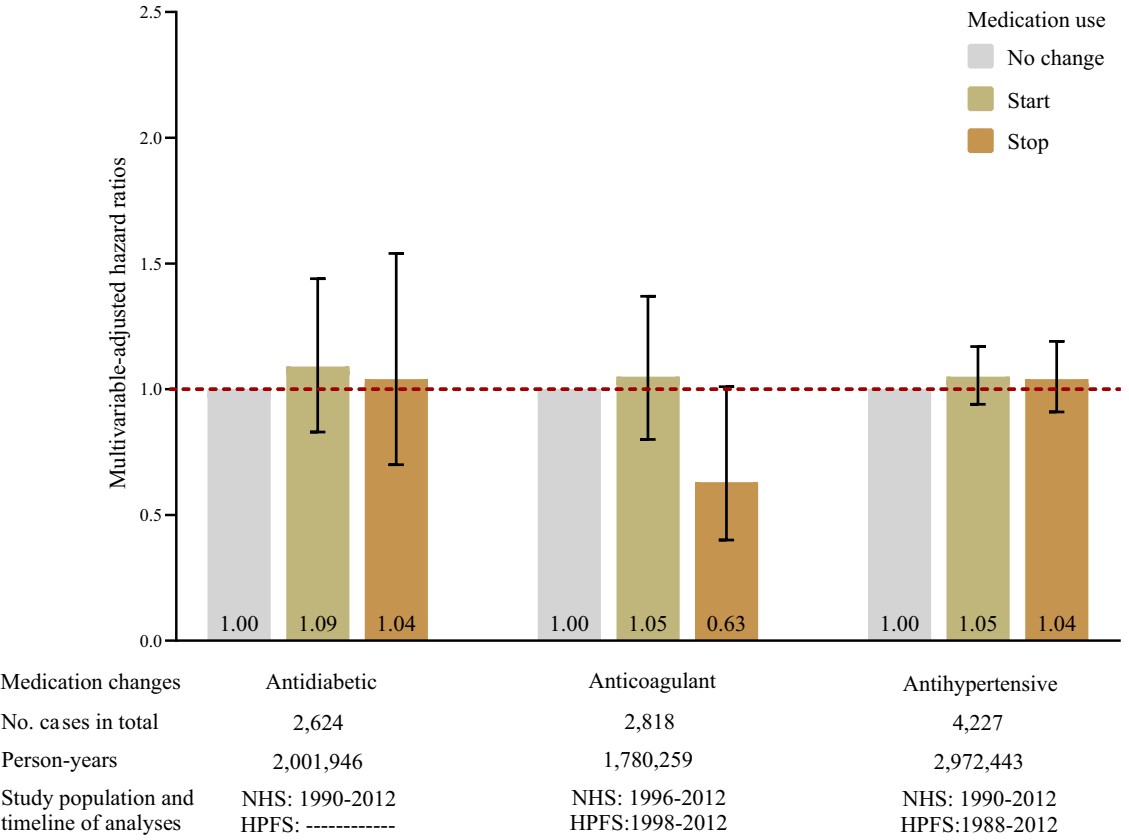

| Medication changes | Antidiabetic | Anticoagulant | Antihypertensive |
|---|---|---|---|
| No. cases in total | 2,624 | 2,818 | 4,227 |
| Person-years | 2,001,946 | 1,780,259 | 2,972,443 |
| Study population and timeline of analyses | NHS: 1990-2012 HPFS: ------------ | NHS: 1996-2012 HPFS:1998-2012 | NHS: 1990-2012 HPFS:1988-2012 |

**Fig. 3 | Recent change in medication use and 2-year risk of other digestive system cancer diagnoses.** Other cancers of the digestive system, including cancers of esophagus, stomach, small intestine, colorectum, anus, liver, and gallbladder. Data are presented as the hazard ratios obtained from Cox proportional hazards models in analyses of antidiabetic ($n = 101,294$ biologically independent participants, 2624 incident other digestive system cancer cases), anticoagulant ($n = 128,718$ biologically independent participants, 2818 incident other digestive system cancer cases), and antihypertensive ($n = 148,973$ biologically independent participants, 4227 incident other digestive system cancer cases) medication changes. Error bars indicate 95% confidence intervals. Multivariable-adjusted hazard ratios were stratified by age (in months), sex/cohort (women, men; in the pooled analyses only), and calendar year of the survey cycle (each 2-year interval); and adjusted for race/ethnicity (white, black, other, unknown), BMI (continuous, kg/m²), physical activity (continuous, MET-hours/week), smoking (continuous, pack-years), alcohol intake (continuous, grams/day), history of diabetes (yes, no), multivitamin use (yes, no), and previous status of use (yes, no). Abbreviations: NHS Nurses' Health Study, HPFS Health Professionals Follow-up Study, BMI body mass index, MET metabolic equivalent task.

prospective population. Our study has several limitations of note. Medication data were assessed via patient report on biennial questionnaires, rather than by medical record review. However, prior studies of medication use in these cohorts have indicated expected associations with relevant outcomes[45-53], and high accuracy of self-report has been identified with multiple lifestyle factors and medical diagnoses[54-60]. We cannot exclude the possibility that some participants changed their medication use after the last questionnaire return date at a median of 12 months prior to clinical cancer diagnosis. However, identifying medication changes very close to the time of cancer diagnosis will be less useful for earlier cancer detection, as the clinical diagnosis was made soon thereafter regardless. Our study participants were healthcare professionals and predominantly of European ancestry, and the sample sizes in subgroup analyses by cohort (sex) were more limited, such that additional studies in both sexes and more diverse populations would be useful. It is unlikely that medication change data alone would have sufficient sensitivity and specificity for risk stratification in the general population. However, these data may provide additive utility in conjunction with other features that can be measured in the EMR or among those with high baseline risk. Lastly, although our study is a large prospective investigation of medication change and pancreatic cancer diagnosis, the number of cases within some medication change categories was relatively modest and additional confirmatory studies are warranted.

In conclusion, in this large prospective study, medication changes are associated with 2-year risk of pancreatic cancer diagnosis. Recent medication changes should be considered as candidate features for incorporation into multi-factor models that identify individuals at near-term risk for pancreatic cancer, though these medication changes are not causally implicated in development of pancreatic cancer.

## Methods

### Ethics statement
This research complies with all relevant ethical regulations. The Institutional Review Boards of the Brigham and Women's Hospital and Harvard T. H. Chan School of Public Health (Boston, Massachusetts) and those of participating registries (as required) approved the study protocols. Written informed consent was obtained from cohort participants to retrieve medical records.

### Study population
The study population included participants from two ongoing large longitudinal cohorts: the NHS and HPFS[17-21]. Briefly, NHS began in 1976[17-20], enrolling 121,700 US female nurses between 30 and 55 years of age. HPFS was initiated in 1986[21], when 51,529 US male health professionals aged 40–75 years were included. In both cohorts, demographics were collected from initial questionnaires at enrollment. Comprehensive information was collected thereafter for

anthropometric measurements, lifestyle characteristics, diet, medical history, and disease outcomes biennially or quadrennially via self-administered follow-up questionnaires.

### Ascertainment of medication use, medication categories, and medication change

Information on medication use was assessed biennially in both cohorts. Participants were asked whether they used specific medications regularly in the past 2 years. Six hypothesis-driven medications categories were investigated (Supplementary Table 1): antidiabetic, anticoagulant, antihypertensive, antacid, NSAID, and antidepressant medications. Throughout follow-up, medication changes were evaluated every 2 years among all participants. For survey cycles in which pancreatic cancer was diagnosed, medication changes in the two-year prediagnosis time window were measured by comparing medication use reported in the questionnaire before diagnosis with the report from the prior questionnaire 2 years earlier (schematic and example demonstrated in Fig. 1). In analyses of antidiabetic medications and antacids, we considered medication escalation by investigating: (1) start of oral hypoglycemic (OHG) medication or insulin, and addition of insulin to OHG medication, and (2) start of proton pump inhibitor (PPI) or histamine H2-receptor ($H_2$) antagonist, and addition of PPI to $H_2$ antagonist.

### Ascertainment of pancreatic cancer cases, other digestive system cancer cases, and participant deaths

Physician-diagnosed incident pancreatic cancer and other digestive system cancer cases were reported by cohort participants via regular questionnaires or identified during follow-up of participant deaths. Medical records and pathology reports were accessed to ascertain diagnoses and tumor characteristics. If medical records were unavailable, cohort investigators referred to state cancer registries. Pancreatic cancer patients with non-adenocarcinoma histology were excluded. Deaths were ascertained through routine searches of the National Death Index, next-of-kin reporting, or postal authorities, with an identification rate exceeding 96%[61,62].

### Ascertainment of covariates

Participant demographics were assessed from returned questionnaires, including age, sex, and race/ethnicity. Participants biennially updated weight, cigarette smoking, physical activity, history of diabetes mellitus, and history of multivitamin use. Dietary data, including information on alcohol intake, were updated quadrennially via validated, semi-quantitative food frequency questionnaires (FFQs). The high validity and reproducibility of information on anthropometrics, lifestyles, diet, and disease diagnoses have been reported[54–60,63–66].

### Statistical analysis

We calculated person-years of follow-up from the return date of the baseline questionnaire until the date of pancreatic cancer diagnosis, death, or follow-up completion (2012 for both cohorts), whichever arrived earliest. Participants with prior history of cancer were excluded at baseline. We also excluded survey cycles with missing medication information reported by participants. In primary analyses, the referent group included those participants who had stable medication use or stable non-use during the two-year period, and the two exposure categories included those participants who started or stopped a medication during this time period. We also performed sensitivity analyses, in which we considered those who had stable use and stable non-use as separate groups (referent group: stable non-use; exposure categories: stable medication use, medication start, and medication stop). Stable refers to medication use or non-use that remained unchanged in the two-year time window, which includes two possible scenarios: stable use and stable non-use of a medication during that time period. We conducted pooled analyses in the combined dataset of NHS and HPFS with assessment of heterogeneity by cohort using random-effects meta-analysis and presented analyses separately (all P-values for heterogeneity by cohort >0.05). The proportional hazards assumption for recent change in medication use was tested using the likelihood ratio test to compare models with and without product terms between each exposure and log-transformed follow-up time, and we detected no violation of the proportional hazards assumption (all P-values for the product terms >0.05). Cox proportional hazards models were used to estimate adjusted hazard ratios (HR) and 95% confidence intervals (95%CI), with age, sex/cohort and calendar year at the beginning of each survey cycle as stratification variables to account for the potential age, cohort and period effects (stratified proportional hazards models assume different baseline hazard functions for different strata). Multivariable-adjusted analyses were stratified by age (in months), sex/cohort (NHS, HPFS), and calendar year of the follow-up survey cycle (each 2-year interval), and adjusted for race/ethnicity (white, black, other, unknown), BMI (continuous, kg/m²), physical activity (continuous, MET-hours/week), smoking (continuous, pack-years), alcohol intake (continuous, grams/day), history of diabetes (yes, no), multivitamin use (yes, no), and baseline status of medication use on preceding questionnaire (yes, no). In sensitivity analyses, we replaced BMI with recent weight change in the two-year prediagnosis time window. In secondary analyses, we explored combined associations of recent changes in medication use and conducted propensity score analysis, given small case numbers within some categories. For propensity score analysis, we estimated the predicted probability of medication change in the two-year prediagnosis time window using logistic regression models that included the above covariates and then adjusted for the propensity score in cox proportional hazards models[67,68]. We also considered incident cases of other digestive system cancer to assess specificity of our findings for pancreatic cancer. Data analyses were performed using SAS statistical software, version 9.4 for UNIX (SAS Institute Inc., Cary, NC). All tests were 2-sided, with P-values < 0.05 indicating statistical significance.

### Reporting summary

Further information on research design is available in the Nature Portfolio Reporting Summary linked to this article.

## Data availability

Data described in the manuscript are available upon formal application to and approval by the Channing Division of Network Medicine at Brigham and Women's Hospital, and Harvard T.H. Chan School of Public Health. To ensure the confidentiality and privacy of cohort participants, written request for access to the data is required. The standard procedure for controlled access requires that applications to use the resources of the Nurses' Health Studies and Health Professionals Follow-up Study undergo a formal review by the cohort committee. The committee assesses the scientific aims, examines the suitability of the proposed methodology for the available data, and confirms that the proposed use aligns with the guidelines of the Ethics and Governance Framework. Further information including the procedures to obtain and access data from the Nurses' Health Study and Health Professionals Follow-Up Study is described at https://www.nurseshealthstudy.org/researchers (email: nhsaccess@channing.harvard.edu) and https://sites.sph.harvard.edu/hpfs/for-collaborators/. Source data are provided with this paper.

## Code availability

The analysis programs are publicly available through https://github.com/drzhangyin/med_change_pdac.

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

## Acknowledgements

The Nurses' Health Study and Health Professionals Follow-up Study were supported by grants UM1 CA186107, U01 CA176726, P01 CA87969, and U01 CA167552 from the National Institutes of Health (NIH). This work was additionally supported by the Swedish Research Council (2021-06739) to Q.L.W; by NIH grant R01 CA205406 and Project P Fund to K.N.; by a grant from the World Cancer Research Fund to E.L.G.; and by Lustgarten Foundation dedicated laboratory program, Dana-Farber Cancer Institute Hale Family Center for Pancreatic Cancer Research, NIH grant U01 CA210171, P50 CA127003, Stand Up to Cancer, Pancreatic Cancer Action Network, Noble Effort Fund, Wexler Family Fund, Promises for Purple, and Bob Parsons Fund to B.M.W. This research was also supported by Stand Up To Cancer-Lustgarten Foundation Pancreatic Cancer Interception Translational Cancer Research grant SU2C-AACR-DT25-17. Y.Z. was supported by Irene M. & Fredrick J. Stare Nutrition Education Fund Doctoral Scholarship and Mayer Fund Doctoral Scholarship. The funding sources played no role in the study design, data collection, data analysis, and interpretation of results, or the decisions made in preparation and submission of the article. The authors thank all participants and staff of the Nurses' Health Studies and the Health Professionals Follow-Up Study for their great contributions to this research. The authors would like to acknowledge the contribution to this study from central cancer registries supported through the Centers for Disease Control and Prevention's National Program of Cancer Registries (NPCR) and/or the National Cancer Institute's Surveillance, Epidemiology, and End Results (SEER) Program. Central registries may also be supported by state agencies, universities, and cancer centers. Participating central cancer registries include the following: Alabama, Alaska, Arizona, Arkansas, California, Colorado, Connecticut, Delaware, Florida, Georgia, Hawaii, Idaho, Indiana, Iowa, Kentucky, Louisiana, Massachusetts, Maine, Maryland, Michigan, Mississippi, Montana, Nebraska, Nevada, New Hampshire, New Jersey, New Mexico, New York, North Carolina, North Dakota, Ohio, Oklahoma, Oregon, Pennsylvania, Puerto Rico, Rhode Island, Seattle SEER Registry, South Carolina, Tennessee, Texas, Utah, Virginia, West Virginia, Wyoming. The authors assume full responsibility for analyses and interpretation of these data.

## Author contributions

Y.Z. and B.M.W. had full access to all the data in the study and take responsibility for the integrity of the data and the accuracy of the data analysis. Concept and design: Y.Z. and B.M.W. Acquisition, analysis, or interpretation of data: Y.Z. Statistical analysis: Y.Z. Drafting of the manuscript: Y.Z. and B.M.W. Critical revision of the manuscript for important intellectual content: All authors. Obtained funding: Y.Z., Q.-L.W., K.N., E.L.G., and B.M.W. Administrative, technical, or material support: All authors. Study supervision: B.M.W.

## Competing interests

K.N. declares institutional research funding from Revolution Medicines, Pharmavite, Evergrande Group, and Janssen outside the submitted work, and consulting/advisory board fees from SeaGen, BiomX, Bicara Therapeutics, X-Biotix Therapeutics, and Redesign Health, outside the submitted work. K.P. declares advisory board fees from Celgene, Eisai, and Helsinn/QED, outside the submitted work. B.M.W. declares research funding from Celgene, Eli Lilly, and Company, Novartis, and Revolution

Medicine, and consulting for Celgene, GRAIL, and Mirati outside the submitted work. Other authors declare no potential conflicts of interest.
