## [Peer Review File · Nature Communications]

Pancreatic cancer is associated with medication changes prior to clinical diagnosisREVIEWER COMMENTS

Reviewer #1 (Remarks to the Author): expertise in pancreatic cancer epidemiology

This is a well written paper addressing a novel method at patient identification for known early signs of pancreatic cancer. This is being done in an attempt to identify patients to be caught at an earlier stage of pancreatic cancer so that interventions may be performed.

It is well established that patients with pancreatic cancer have antecedent diabetes in the months to years prior to the diagnosis of cancer. There are also other metabolic changes such as weight loss and changes in diet that can impact need for antihypertensive medications. This study takes advantage of data from 2 prospective cohorts, the Nurses' Health Study and the Health Professionals Follow-up Study to examine whether specific medication changes are associated with increased risk for imminent diagnosis of pancreatic cancer

Included in the study are nearly 150,000 participants and 991 incident pancreatic cancer cases during the prospective follow-up. There are nearly three million person-years of follow up. A high proportion of cases are validated and covariables include sex, race, BMI, physical activity, smoking, alcohol intake, history of diabetes, and multivitamin use. These are appropriate although family history is not included.

As expected, initiation of anti-diabetic therapy, whether an oral hypoglycemic or insulin, is associated with increased risk for pancreatic cancer within the next two years. In addition, discontinuation of an antihypertensive is also associated with risk for pancreatic cancer. Initiation of an anticoagulant does not quite meet the threshold for statistical significance. The combined effect of initiation of antidiabetic therapy and discontinuation of antihypertensive medication use carries a high relative risk of with a hazard ratio of nearly 5. This is the crux of the conclusion of the article.

The authors rightly conclude that use of electronic medical records carries the possibility of identifying discrete changes in medical care including changes in medications possibly with the ability to flag patterns such as this and identify patients at higher risk.

My main concern about this article is not the validity of the findings or the methodology. I think the study has been well done and it is likely valid. My main concern is in the actual clinical impact of this finding. It does provide a discreet identifiable way to measure a new diagnosis of diabetes and potential causes of decreased need for antihypertensives including weight loss or changes in nutritional state. However, there are other potential ways to identify these changes that may be perhaps more sensitive.

In looking at figure 3, which includes patients who have at least two medication changes such as start of antidiabetic or anticoagulant medication and stop event hypertensive medications, there were only 6 patients who had at least two changes. In Table 1, when identifying patients who had a start of antidiabetic therapy and stoppage of an anti-hypertensive, there were only 4 patients who developed pancreatic cancer. So, using this methodology of focusing on patients who start anti diabetic therapy and stop antihypertensive therapy, the study would have identified 4 out of 991 patients. Only 33 patients of the 991 had start of an anti-diabetic noted in the two years prior to diagnosis according to table one. I realize not all of the patients who were diagnosed were included in Table 1, possibly for absence of covariates etc, but it nonetheless is clear that it is a small minority of patients that would be identified with alerts for these changes.

I do think it would be helpful to give an absolute risk for pancreatic cancer for patients identified with these changes. For instance, it is estimated that approximately one out of 125 new diabetics over age 50 will develop pancreatic cancer in the next couple of years from other published studies. A similar absolute risk would help this manuscript to enable clinicians to then make a judgment call when they see this pattern. Even if electronic alerts such as a best practice advisory were to be developed, an absolute risk would be helpful. This could then help justify either screening certain populations of individuals given an absolute risk or enable other inputs such as germline genetics or circulating molecular markers to most accurately identify patients in need of imaging or other types of workup.

Robert McWilliams, MD, MSc

Reviewer #2 (Remarks to the Author): expertise in risk factor modelling and biostatistics

This welcome report aims to fulfil an important research gap by identifying additional clinical factors that could inform risk prediction models for pancreatic cancer. This is a timely topic; a systematic review on clinical factors for pancreatic cancer has just been published and medications were an understudied component of these.

This article is based on analysis from the robust Nurses' Health Study and Health Professional Follow-Up study cohorts in the US. Medication changes in the two years prior to a pancreatic cancer diagnosis were explored; starting antidiabetic medications and ceasing antihypertensive medications were most strongly associated with pancreatic cancer development. This is consistent with knowledge that new-onset diabetes is associated with pancreatic cancer development. The report is well written. Investigation of medication windows 2 years prior to diagnosis is an arguably narrow window, however not unreasonable for early detection purposes. Some comments for consideration are below.

1. The overall results appear to be stronger and/or limited to findings from the Nurses Health Study cohort (antidiabetic medications weren't reported in the HPFUS/Supp Table 7, while HRs for the antihypertensive medications were stronger in the NHS cohort than NHS/HPFUS combined). Is it possible that these results are really only reflective of known prediction factors in women?
2. The overall sample size is large (991 pancreatic cancer cases) but the number of medication initiators and stoppers is relatively small in many analyses. While I appreciate the visual presentation of results in Figures 2,3 and 4, the Supplementary Table 4 is actually the most informative and I would encourage that it is promoted to a main table in the manuscript.
3. Following on from the above comment, the relatively small number of medication starters/stoppers in some analyses should be added as a limitation in the discussion (important to note for editors that this is still a large study in the context of pancreatic cancer)
4. Results lines 230-232 description of stable non-use and Supplementary Table 5 results could be explained more clearly to understand how this analysis differs from the main analysis.
5. The rationale for studying stopping antihypertensive medications is hypothesised to be via weight loss and subsequent reductions in blood pressure; is there any evidence behind this? This was the only medication category of 'stopping use' that was investigated so it is important to understand the rationale for this.
6. Please add sex/cohort to Supp Tables 2 and 3 to make it more obvious which cohorts were studied in each analysis.
7. History of diabetes was included as a confounder in some multivariate analysis, I wasn't sure if this was appropriate for antidiabetic medications, is this not a mediator/on the causal pathway? Consider the addition of a DAG figure.

RESPONSE LETTER TO REVIEWER COMMENTS

Please note that all page and paragraph numbers refer to the tracked change version of the revised manuscript uploaded as “Related Manuscript File”.

Response to Reviewer 1 (expert in pancreatic cancer epidemiology)’s comments

1. This is a well written paper addressing a novel method at patient identification for known early signs of pancreatic cancer. This is being done in an attempt to identify patients to be caught at an earlier stage of pancreatic cancer so that interventions may be performed.

It is well established that patients with pancreatic cancer have antecedent diabetes in the months to years prior to the diagnosis of cancer. There are also other metabolic changes such as weight loss and changes in diet that can impact need for antihypertensive medications. This study takes advantage of data from 2 prospective cohorts, the Nurses’ Health Study and the Health Professionals Follow-up Study to examine whether specific medication changes are associated with increased risk for imminent diagnosis of pancreatic cancer.

Included in the study are nearly 150,000 participants and 991 incident pancreatic cancer cases during the prospective follow-up. There are nearly three million person-years of follow up. A high proportion of cases are validated and covariables include sex, race, BMI, physical activity, smoking, alcohol intake, history of diabetes, and multivitamin use. These are appropriate although family history is not included.

As expected, initiation of anti-diabetic therapy, whether an oral hypoglycemic or insulin, is associated with increased risk for pancreatic cancer within the next two years. In addition, discontinuation of an antihypertensive is also associated with risk for pancreatic cancer. Initiation of an anticoagulant does not quite meet the threshold for statistical significance. The combined effect of initiation of antidiabetic therapy and discontinuation of antihypertensive medication use carries a high relative risk of with a hazard ratio of nearly 5. This is the crux of the conclusion of the article.

The authors rightly conclude that use of electronic medical records carries the possibility of identifying discrete changes in medical care including changes in medications possibly with the ability to flag patterns such as this and identify patients at higher risk.

Response: We thank the reviewer for the favorable consideration of our manuscript and very helpful suggestions below. We have responded to them point by point and made further revisions to our manuscript.

2. A high proportion of cases are validated and covariables include sex, race, BMI, physical activity, smoking, alcohol intake, history of diabetes, and multivitamin use. These are appropriate although family history is not included.

Response: We appreciate the reviewer’s comment. We agree with the reviewer that rigorous control for potential confounding has been considered in our multivariable-adjusted analyses. For the reviewer’s further consideration, we have now performed an additional sensitivity analysis by also adjusting for family history of cancer, which was not previously included in our multivariable-adjusted models. We have provided the Editor/Reviewer Table 1 below. As evident from this table, the effect estimates remained essentially unchanged after additionally adjusting for family history of cancer. We would be willing to incorporate these results into the revised manuscript if this is deemed necessary by the editors or reviewers but have not done so currently in the revised manuscript due to the highly similar effect estimates after inclusion of family history in multivariable-adjusted models.

Editor/Reviewer Table 1. Recent change in medication use and 2-year risk of pancreatic cancer diagnosis in the pooled NHS and HPFS cohorts

	Recent change in medication use ^{a,b}		
	No change	Start	Stop
Antidiabetic medications			
No. of cases	538	33	17
Person-years	1,966,617	33,435	15,660
MV-adjusted ^c	1	1.99 (1.30-3.03)	1.55 (0.86-2.81)
MV-adjusted ^d (additionally adjusted for family history of cancer)	1	1.99 (1.30-3.03)	1.55 (0.86-2.81)
Anticoagulant medications			
No. of cases	658	20	10
Person-years	1,754,044	25,461	12,227
MV-adjusted ^c	1	1.50 (0.95-2.35)	1.18 (0.57-2.46)
MV-adjusted ^d (additionally adjusted for family history of cancer)	1	1.50 (0.96-2.36)	1.19 (0.57-2.48)
Antihypertensive medications			
No. of cases	789	86	116
Person-years	2,627,927	224,273	141,857
MV-adjusted ^c	1	1.03 (0.81-1.30)	1.77 (1.42-2.20)
MV-adjusted ^d (additionally adjusted for family history of cancer)	1	1.02 (0.81-1.29)	1.77 (1.42-2.21)
Antacids			
No. of cases	229	21	19
Person-years	539,960	62,898	45,475
MV-adjusted ^c	1	0.76 (0.48-1.21)	1.06 (0.58-1.93)
MV-adjusted ^d (additionally adjusted for family history of cancer)	1	0.76 (0.48-1.21)	1.06 (0.58-1.93)
NSAIDs			
No. of cases	800	49	54
Person-years	2,333,006	148,689	129,881
MV-adjusted ^c	1	1.04 (0.76-1.42)	1.14 (0.85-1.52)
MV-adjusted ^d (additionally adjusted for family history of cancer)	1	1.04 (0.76-1.42)	1.14 (0.85-1.52)
Antidepressants			
No. of cases	707	20	28
Person-years	1,790,806	51,326	38,441
MV-adjusted ^c	1	0.99 (0.63-1.55)	1.39 (0.85-2.27)
MV-adjusted ^d (additionally adjust for family history of cancer)	1	0.99 (0.63-1.55)	1.39 (0.85-2.27)

Abbreviations: NHS, Nurses' Health Study; HPFS, Health Professionals Follow-up Study; MV, multivariate; BMI, body mass index; MET, metabolic equivalent task; NSAIDs, non-steroidal anti-inflammatory drugs.

^a Follow-up time: antidiabetic medications (NHS: 1990-2010; information on antidiabetic medication use in HPFS was assessed relatively late (in 2008 and afterwards) and was therefore not included in the analyses),

anticoagulant (NHS: 1996-2012; HPFS: 1998-2012), antihypertensive medications (NHS: 1990-2012; HPFS: 1988-2012), antacids (NHS: 2002-2012; HPFS: 2006-2012), NSAIDs (NHS: 1992-2012; HPFS: 1988-2012), and antidepressants (NHS: 1998-2012; HPFS: 1992-2012).

^b Medication change was measured by comparing questionnaires returned at a median of 1 year (current status of use) and 3 years (previous status of use) before pancreatic cancer diagnosis.

^c Stratified by age (in months), sex/cohort (women, men; in the pooled analyses only), and calendar year of the survey cycle (each 2-year interval); adjusted for race (white, black, other, unknown), BMI (continuous, kg/m²), physical activity (continuous, MET-hours/week), smoking (continuous, pack-years), alcohol intake (continuous, grams/day), history of diabetes (yes, no), multivitamin use (yes, no), and previous status of use (yes, no).

^d Additionally adjust for family history of cancer.

3. My main concern about this article is not the validity of the findings or the methodology. I think the study has been well done and it is likely valid. My main concern is in the actual clinical impact of this finding. It does provide a discreet identifiable way to measure a new diagnosis of diabetes and potential causes of decreased need for antihypertensives including weight loss or changes in nutritional state. However, there are other potential ways to identify these changes that may be perhaps more sensitive.

Response: We appreciate the reviewer's comment. In this large prospective study, initiation of antidiabetic and cessation of antihypertensive medications were both associated with 2-year risk of developing pancreatic cancer. It is noteworthy that these risks persisted even when other lifestyle factors and comorbidities (such as obesity, diabetes, and physical activity) were comprehensively controlled for in multivariable-adjusted models. Considering few individual exposures portend sufficiently high pancreatic cancer risk to justify imaging surveillance, we think that identification of predictive features such as patterns of medication changes will be important to multi-factor models that can identify those with high near-term pancreatic cancer risk in the general population.

We agree with the reviewer that there are potentially other ways to identify new-onset metabolic alterations (e.g., hyperglycemia and weight loss) and other symptoms or conditions (e.g., venous thromboembolism) in clinical settings. However, as we discussed in the Discussion section of the manuscript, outpatient prescriptions have become searchable structured fields within the modern EMR and changes to medication prescribing can be a highly efficient way to assess new disease states from structured data, avoiding complexities associated with billing codes and free text notes. Furthermore, changes in medication use can inform disease severity over time, providing more information than the simple presence or absence of a condition. Given the increasing availability of structured data on medication use in the EMR and its potential to provide utility beyond available disease codes, we conducted the current study as a proof of concept examination of whether changes in medication use could inform 2-year risk of a pancreatic cancer diagnosis. Medication changes were independently associated with 2-year risk of pancreatic cancer diagnosis in our study, even when substantial information was available for other lifestyle factors and comorbid conditions, such as might be available in the EMR. Thus, these results suggest that patterns of medication changes should be considered for inclusion in multi-factor models that identify high-risk individuals for pancreatic cancer even when other data types are available from the EMR.

4. In looking at figure 3, which includes patients who have at least two medication changes such as start of antidiabetic or anticoagulant medication and stop event hypertensive medications, there were only 6 patients who had at least two changes. In Table 1, when identifying patients who had a start of antidiabetic therapy and stoppage of an anti-hypertensive, there were only 4 patients who developed pancreatic cancer. So, using this methodology of focusing on patients who start anti diabetic therapy and stop antihypertensive therapy, the study would have identified 4 out of 991 patients. Only 33 patients of the 991 had start of an anti-diabetic noted in the two years prior to diagnosis according to table one. I realize not all of the patients who were diagnosed were included in Table 1, possibly for absence of covariates etc, but it nonetheless is

clear that it is a small minority of patients that would be identified with alerts for these changes. I do think it would be helpful to give an absolute risk for pancreatic cancer for patients identified with these changes. For instance, it is estimated that approximately one out of 125 new diabetics over age 50 will develop pancreatic cancer in the next couple of years from other published studies. A similar absolute risk would help this manuscript to enable clinicians to then make a judgment call when they see this pattern. Even if electronic alerts such as a best practice advisory were to be developed, an absolute risk would be helpful. This could then help justify either screening certain populations of individuals given an absolute risk or enable other inputs such as germline genetics or circulating molecular markers to most accurately identify patients in need of imaging or other types of workup.

Response: We thank the reviewer for this comment. We agree with the reviewer that it would be helpful to provide readers with additional results on absolute risk, and we have now done so in the tables and supplementary tables within the revised manuscript. Specifically, we provide information on incidence rate per 100,000 person-years in all tables and supplementary tables throughout the revised manuscript.

Response to Reviewer 2 (expert in risk factor modelling and biostatistics)'s comments

1. This welcome report aims to fulfil an important research gap by identifying additional clinical factors that could inform risk prediction models for pancreatic cancer. This is a timely topic; a systematic review on clinical factors for pancreatic cancer has just been published and medications were an understudied component of these.

Response: We thank the reviewer for the positive comments and helpful suggestions below. We have addressed the reviewer's comments point by point and made the appropriate edits in the revised manuscript. In the revised manuscript, we have also now added a citation for a recent systematic review on pancreatic cancer risk prediction models (Ralph Santos, et al. Clinical Prediction Models for Pancreatic Cancer in General and At-Risk Populations: A Systematic Review. *Am J Gastroenterol.* 2022. PMID: 36148840 [Online ahead of print]).

2. This article is based on analysis from the robust Nurses' Health Study and Health Professional Follow-Up study cohorts in the US. Medication changes in the two years prior to a pancreatic cancer diagnosis were explored; starting antidiabetic medications and ceasing antihypertensive medications were most strongly associated with pancreatic cancer development. This is consistent with knowledge that new-onset diabetes is associated with pancreatic cancer development. The report is well written. Investigation of medication windows 2 years prior to diagnosis is an arguably narrow window, however not unreasonable for early detection purposes.

Response: We thank the reviewer for the positive comments. Yes, we considered the 2-year time window in our analysis based on clinical observations that patients with pancreatic cancer commonly develop the new onset of symptoms and signs in the 1-2 years before their cancer is diagnosed. These changes in symptoms and signs can lead individuals to change to their patterns of medication use in this time window and may therefore show predictive value in defining those at higher risk of a subsequent pancreatic cancer diagnosis.

3. The overall results appear to be stronger and/or limited to findings from the Nurses Health Study cohort (antidiabetic medications weren't reported in the HPFUS/Supp Table 7, while HRs for the antihypertensive medications were stronger in the NHS cohort than NHS/HPFUS combined). Is it possible that these results are really only reflective of known prediction factors in women?

Response: We thank the reviewer for this comment. We agree with the reviewer that the effect estimates in the NHS cohort were generally numerically stronger compared to those in the HPFS cohort. However,

the HPFS cohort had a smaller sample size with fewer pancreatic cancer cases, and all *P* values for heterogeneity by cohort were >0.05, indicating that statistically significant heterogeneity between the NHS and HPFS cohorts was not identified. Therefore, based on the current results, we could not draw the conclusion that these findings applied only to women, and further studies are warranted to validate our findings in additional populations of both sexes.

To address the reviewer's comment, we have edited the text in the Discussion section of the revised manuscript on page 6 (paragraph 1), where we now state:

“Our study participants were healthcare professionals and predominantly of European ancestry, and the sample sizes in subgroup analyses by cohort (sex) were more limited, such that additional studies in both sexes and more diverse populations would be useful.”

4. The overall sample size is large (991 pancreatic cancer cases) but the number of medication initiators and stoppers is relatively small in many analyses. While I appreciate the visual presentation of results in Figures 2,3 and 4, the Supplementary Table 4 is actually the most informative and I would encourage that it is promoted to a main table in the manuscript.

Response: We thank the reviewer for this suggestion. We agree with the reviewer that Supplementary Table 4 is highly informative, and we have now promoted it to a main table in the revised manuscript (Table 1 in the revised manuscript), with minor edits for brevity. Given this change, we have removed Figure 2 (visual presentation of Supplementary Table 4) from the revised manuscript.

In response to the reviewer's comment, we have now included the below table as Table 1 of the revised manuscript:

Table 1. Recent change in medication use and 2-year risk of pancreatic cancer diagnosis in the pooled NHS and HPFS cohorts

	Recent change in medication use ^{a,b}		
	No change	Start	Stop
Any antidiabetic medications			
No. of cases	538	33	17
Person-years	1,966,617	33,435	15,660
Incidence rate (per 100,000 person-years)	27 (25-30)	99 (70-139)	109 (67-175)
Age-adjusted ^c	1	2.99 (2.10-4.27)	2.76 (1.70-4.50)
MV-adjusted ^d	1	1.99 (1.30-3.03)	1.55 (0.86-2.81)
Insulin			
No. of cases	570	13	5
Person-years	2,002,769	8,653	4,290
Incidence rate (per 100,000 person-years)	28 (26-31)	150 (87-259)	117 (49-280)
Age-adjusted ^c	1	4.69 (2.69-8.18)	2.83 (1.17-6.87)
MV-adjusted ^c	1	2.84 (1.58-5.12)	1.18 (0.43-3.24)
Anticoagulant medications			
No. of cases	658	20	10
Person-years	1,754,044	25,461	12,227
Incidence rate (per 100,000 person-years)	38 (35-40)	79 (51-122)	82 (44-152)
Age-adjusted ^c	1	1.60 (1.02-2.52)	1.52 (0.81-2.87)
MV-adjusted ^d	1	1.50 (0.95-2.35)	1.18 (0.57-2.46)

Antihypertensive medications

No. of cases	789	86	116
Person-years	2,627,927	224,273	141,857
Incidence rate (per 100,000 person-years)	30 (28-32)	38 (31-47)	82 (68-98)
Age-adjusted ^c	1	1.04 (0.83-1.31)	1.95 (1.60-2.38)
MV-adjusted ^d	1	1.03 (0.81-1.30)	1.77 (1.42-2.20)

Antacids

No. of cases	229	21	19
Person-years	539,960	62,898	45,475
Incidence rate (per 100,000 person-years)	42 (37-48)	33 (22-51)	42 (27-66)
Age-adjusted ^c	1	0.81 (0.52-1.28)	1.01 (0.63-1.62)
MV-adjusted ^d	1	0.76 (0.48-1.21)	1.06 (0.58-1.93)

NSAIDs

No. of cases	800	49	54
Person-years	2,333,006	148,689	129,881
Incidence rate (per 100,000 person-years)	34 (32-37)	33 (25-44)	42 (32-54)
Age-adjusted ^c	1	0.96 (0.71-1.29)	1.17 (0.88-1.55)
MV-adjusted ^d	1	1.04 (0.76-1.42)	1.14 (0.85-1.52)

Antidepressants

No. of cases	707	20	28
Person-years	1,790,806	51,326	38,441
Incidence rate (per 100,000 person-years)	39 (37-42)	39 (25-60)	73 (50-105)
Age-adjusted ^c	1	1.05 (0.67-1.65)	1.88 (1.28-2.75)
MV-adjusted ^d	1	0.99 (0.63-1.55)	1.39 (0.85-2.27)

Abbreviations: NHS, Nurses' Health Study; HPFS, Health Professionals Follow-up Study; MV, multivariate; BMI, body mass index; MET, metabolic equivalent task; NSAIDs, non-steroidal anti-inflammatory drugs.

^a Follow-up time: antidiabetic medications (NHS: 1990-2010; information on antidiabetic medication use in HPFS was assessed relatively late (in 2008 and afterwards) and was therefore not included in the analyses), anticoagulant (NHS: 1996-2012; HPFS: 1998-2012), antihypertensive medications (NHS: 1990-2012; HPFS: 1988-2012), antacids (NHS: 2002-2012; HPFS: 2006-2012), NSAIDs (NHS: 1992-2012; HPFS: 1988-2012), and antidepressants (NHS: 1998-2012; HPFS: 1992-2012).

^b Medication change was measured by comparing questionnaires returned at a median of 1 year (current status of use) and 3 years (previous status of use) before pancreatic cancer diagnosis.

^c Stratified by age (in months), sex/cohort (women, men; in the pooled analyses only), and calendar year of the survey cycle (each 2-year interval).

^d Stratified by age (in months), sex/cohort (women, men; in the pooled analyses only), and calendar year of the survey cycle (each 2-year interval); adjusted for race (white, black, other, unknown), BMI (continuous, kg/m²), physical activity (continuous, MET-hours/week), smoking (continuous, pack-years), alcohol intake (continuous, grams/day), history of diabetes (yes, no), multivitamin use (yes, no), and previous status of use (yes, no).

5. Following on from the above comment, the relatively small number of medication starters/stoppers in some analyses should be added as a limitation in the discussion (important to note for editors that this is still a large study in the context of pancreatic cancer)

Response: We thank the reviewer for this comment and have now acknowledged this as a potential limitation in our revised manuscript.

Within the Discussion section of the revised manuscript on page 6 (paragraph 1), we now state:

“Lastly, although our study is a large prospective investigation of medication change and pancreatic cancer diagnosis, the number of cases within some medication change categories was relatively modest and additional confirmatory studies are warranted.”

6. Results lines 230-232 description of stable non-use and Supplementary Table 5 results could be explained more clearly to understand how this analysis differs from the main analysis.

Response: We thank the reviewer for this comment. In our primary analysis, the referent group included those participants who had stable medication use or non-use during the two-year period, and the two exposure categories included those participants who started or stopped a medication. For these analyses, “stable” refers to medication use or non-use that remained unchanged in the two-year time window. In the sensitivity analysis, we considered those who had stable medication use and stable non-use during the two-year period as separate groups. Specifically, we used those with stable non-use of a medication class as the referent group, and the three exposure categories included stable medication use, medication start, and medication stop.

In response to the reviewer’s suggestion, we have now added more detail to the manuscript text and the footnote of Supplementary Table 4 in our revised manuscript.

In the Statistical Analysis section of the revised manuscript on page 7 (paragraph 2), we now state:

“In primary analyses, the referent group included those participants who had stable medication use or stable non-use during the two-year period, and the two exposure categories included those participants who started or stopped a medication during this time period. We also performed sensitivity analyses, in which we considered those who had stable use and stable non-use as separate groups (referent group: stable non-use; exposure categories: stable medication use, medication start, and medication stop). “Stable” refers to medication use or non-use that remained unchanged in the two-year time window, which includes two possible scenarios: stable use and stable non-use of a medication during that time period.”

In the footnote of Supplementary Table 5 of the revised manuscript, we also now state:

““Stable” refers to medication use or non-use that remained unchanged in the two-year time window, which includes two possible scenarios: stable use and stable non-use over the two years.”

7. The rationale for studying stopping antihypertensive medications is hypothesised to be via weight loss and subsequent reductions in blood pressure; is there any evidence behind this? This was the only medication category of 'stopping use' that was investigated so it is important to understand the rationale for this.

Response: We thank the reviewer for this question. As we mentioned in the Introduction and Discussion and sections of the manuscript, weight loss commonly occurs in patients who develop pancreatic cancer, and this weight loss is often seen within the 1 year prior to pancreatic cancer diagnosis. A host of other metabolic disturbances have also been identified prior to diagnosis in these patients, including hyperglycemia, decreased serum lipids, increased body temperature, and decreased skeletal muscle and adipose tissue. In clinical practice, we commonly see decreased patient blood pressure during the same prediagnosis time window as weight loss and metabolic changes, suggesting the potential utility of measuring reduction or cessation of blood pressure medications as a marker of an upcoming pancreatic cancer diagnosis. Given these prior data and clinical observations, we hypothesized that stopping antihypertensive medications may have utility in risk stratification for pancreatic cancer. In response to the

reviewer's comment, we have ensured citations to the below references are included in the revised manuscript.

References

1. Yuan C, et al. Diabetes, Weight Change, and Pancreatic Cancer Risk. *JAMA Oncol* 6, e202948 (2020).
2. Sah RP, et al. Phases of Metabolic and Soft Tissue Changes in Months Preceding a Diagnosis of Pancreatic Ductal Adenocarcinoma. *Gastroenterology* 156, 1742-1752 (2019).
3. Khalaf N, Wolpin BM. Metabolic Alterations as a Signpost to Early Pancreatic Cancer. *Gastroenterology* 156, 1560-1563 (2019).
4. Huang BZ, et al. New-Onset Diabetes, Longitudinal Trends in Metabolic Markers, and Risk of Pancreatic Cancer in a Heterogeneous Population. *Clin Gastroenterol Hepatol* 18, 1812-1821 e1817 (2020).
5. Sharma A, et al. Model to Determine Risk of Pancreatic Cancer in Patients With New-Onset Diabetes. *Gastroenterology* 155, 730-739 e733 (2018).

8. Please add sex/cohort to Supp Tables 2 and 3 to make it more obvious which cohorts were studied in each analysis.

Response: We agree with the reviewer that information on which cohorts were studied for each analysis can be made more explicit to further improve readability in these two supplementary tables. In response to this suggestion, we have now added sex/cohort information to Supplementary Tables 2 and 3 for each analysis.

9. History of diabetes was included as a confounder in some multivariate analysis, I wasn't sure if this was appropriate for antidiabetic medications, is this not a mediator/on the causal pathway? Consider the addition of a DAG figure.

Response: We thank the reviewer for this comment. We agree with the reviewer that a confounder should have three key properties: 1) serve as a risk factor for the outcome; 2) be associated with the exposure in the participant population; and 3) not exist on the causal pathway between exposure and outcome or as a consequence of the outcome. In this case, history of diabetes is associated with both exposure (medication change) and outcome (pancreatic cancer). However, the relationships are somewhat complex, since diabetes can act as both a "causal" risk factor for pancreatic cancer and as a consequence of the undiagnosed cancer. Prior studies also suggest that a diagnosis of diabetes is not included in a patient's chart for months to years after elevated glucose is identified (PMIDs: 17149994, 24705614, 30175870, etc.). Furthermore, medication changes can be made due to worsening glucose control, even when a diagnosis of diabetes was previously established. Thus, we hypothesized that medication changes might serve as a marker for the new-onset of hyperglycemia and/or the worsening glucose control commonly seen prior to a pancreatic cancer diagnosis, providing additional information beyond the binary exposure of whether diabetes had been previously identified. To test this hypothesis, we opted to include diabetes history in our multivariable-adjusted models to examine whether medication changes provided additional risk information even when diabetes diagnosis information was available. However, to ensure readers had access to all study results, we included in Supplementary Tables 4 and 5 of the revised manuscript: age-adjusted models, models further adjusted for body-mass index and diabetes alone, and models fully adjusted for all covariates. With this approach, we looked to provide readers with information on how relative risks changed with adjustment for diabetes compared to a limited age-adjusted model and a fully-adjusted model. Overall, we would advocate that our results suggest additional predictive information was present for medication change data beyond that contained in the binary diabetes history variable. We hope that the reviewer finds this rationale compelling for our decision to include diabetes in multivariable-adjusted models examining diabetes medication changes, although we are happy to include additional models if the reviewer would find them useful.

REVIEWER COMMENTS

Reviewer #1 (Remarks to the Author):

All criticisms satisfactorily addressed, no further comments

Reviewer #2 (Remarks to the Author):

The response letter to previous comments is exemplary, and thoroughly addresses all suggestions made for manuscript changes including edits to tables/figures and inclusion of additional references where requested. The resulting manuscript is an important addition to the literature in this field.

RESPONSE TO REVIEWER COMMENTS

Reviewer #1:

All criticisms satisfactorily addressed, no further comments

Response: We appreciate the Reviewer's comment that we have addressed all criticisms satisfactorily and no further comments remain.

Reviewer #2:

The response letter to previous comments is exemplary, and thoroughly addresses all suggestions made for manuscript changes including edits to tables/figures and inclusion of additional references where requested. The resulting manuscript is an important addition to the literature in this field.

Response: We appreciate the Reviewer's comment that our prior response letter was "exemplary" and "thoroughly addresses all suggestions made for manuscript changes".